# Chinese Medicine-Derived Natural Compounds and Intestinal Regeneration: Mechanisms and Experimental Evidence

**DOI:** 10.3390/biom15091212

**Published:** 2025-08-22

**Authors:** Fengbiao Guo, Shaoyi Zhang

**Affiliations:** 1Institute of Chinese Medical Sciences, State Key Laboratory of Mechanism and Quality of Chinese Medicine, University of Macau, Macau SAR 999078, China; fengbiaoguo@um.edu.mo; 2Department of Pharmaceutical Sciences, Faculty of Health Sciences, University of Macau, Macau SAR 999078, China

**Keywords:** intestinal stem cell, herbal medicine, regenerative pharmacology, microbiome metabolite axis

## Abstract

Intestinal regeneration is essential for maintaining epithelial integrity and repairing mucosal damage caused by inflammation, infections, or injuries. Traditional Chinese Medicine (TCM) has long utilized herbal remedies for gastrointestinal disorders, and accumulating evidence highlights that natural compounds derived from TCM possess significant regenerative potential. This review summarizes the multifaceted mechanisms by which these bioactive compounds promote intestinal healing. Key actions include the stimulation of intestinal stem cell (ISC) proliferation and differentiation, the modulation of inflammatory responses, the reinforcement of epithelial barrier integrity, the attenuation of oxidative stress, and the reshaping of the gut microbiota. Representative compounds such as Astragalus polysaccharides, berberine, curcumin, puerarin, and flavonoids like quercetin exhibit these effects through signaling pathways, including HIF-1, Wnt/β-catenin, NF-κB, Nrf2, and IL-22. Evidence from in vitro organoid models and in vivo studies in colitis, radiation injury, antibiotic-associated diarrhea, and intestinal dysmotility and diarrhea models demonstrates that these compounds enhance crypt villus regeneration, preserve tight junctions, and improve clinical outcomes. The holistic, multi-target actions of Chinese medicine-derived natural products make them promising candidates for therapeutic strategies aimed at intestinal repair. Further clinical validation and mechanistic studies are warranted to facilitate their integration into modern gastrointestinal medicine.

## 1. Introduction

The intestinal epithelium is one of the most rapidly renewing tissues in the human body, capable of complete self-renewal roughly every 3–5 days. This continuous regeneration is driven by intestinal stem cells (ISCs) residing at the base of crypts, which proliferate and differentiate to replenish various epithelial cell lineages [1]. Efficient intestinal regeneration is critical for maintaining gut homeostasis and for healing after an injury or inflammation. In conditions such as inflammatory bowel disease (IBD), an acute injury (e.g., radiation exposure), an infection, or other stressors, normal regenerative processes may be impaired, leading to mucosal damage, barrier dysfunction, and illness. Enhancing the regenerative capacity of the intestine—for instance, by protecting or stimulating ISCs, modulating the immune response, and improving the gut microenvironment—is therefore an important therapeutic goal in gastrointestinal medicine [2,3,4].

Traditional Chinese Medicine (TCM) has long employed herbal remedies and natural products for gastrointestinal ailments, and many of these remedies are now recognized to contain bioactive compounds with potent biological effects. In recent years, there has been growing scientific interest in Chinese medicine derived natural compounds as promoters of intestinal repair and regeneration. Many plant-derived molecules (e.g., alkaloids, flavonoids, polysaccharides, etc.) and herbal formulations have demonstrated the ability to enhance the healing of the intestinal mucosa in experimental models. These compounds often act via multiple converging mechanisms—such as stimulating intestinal stem cell activity, regulating inflammation, strengthening the epithelial barrier, reducing oxidative stress, and even reshaping the gut microbiota [5,6]. Such multi-targeted actions align well with the holistic approach of TCM and suggest that these natural compounds could be valuable adjuncts or alternatives to conventional therapies for disorders involving impaired intestinal regeneration (for example, IBD or intestinal radiation injury).

In this review, we provide a comprehensive overview of how Chinese medicine-derived natural compounds influence intestinal regeneration. We discuss a broad range of mechanisms by which these compounds exert their effects, including the modulation of intestinal stem cells, the regulation of inflammation, the improvement of the epithelial barrier function, the reduction in oxidative stress, and interactions with the gut microbiota. We highlight evidence from both in vitro studies (including intestinal organoid models) and in vivo experiments that illustrate these mechanisms. Through detailed examples from peer-reviewed studies, we aim to delineate the pharmacological actions of these natural compounds and their potential therapeutic implications for promoting intestinal mucosal healing.

## 2. Mechanisms of Action of Natural Compounds in Intestinal Regeneration

### 2.1. Modulation of Intestinal Stem Cells and Epithelial Lineage Differentiation

A fundamental aspect of intestinal regeneration is the activation and differentiation of ISCs to replace the damaged epithelium. Various natural compounds from Chinese medicinal herbs have demonstrated the ability to support ISCs’ survival, proliferation, and lineage commitment. For example, Astragalus polysaccharide (APS) (Figure 1a)—a bioactive polysaccharide from *Astragalus membranaceus* (Huangqi)—has been shown to robustly promote ISC-driven regeneration in injured intestines. In a recent study of radiation-induced intestinal injury, an APS treatment improved the survival of irradiated mice and markedly enhanced crypt regeneration, evidenced by increased numbers of ISCs and crypt survival, along with an elevated expression of ISC markers (e.g., Lgr5) and tight junction proteins in the intestine. Intestinal organoid assays confirmed that APS directly promotes the regeneration of ISCs ex vivo. Mechanistically, APS was found to activate the hypoxia-inducible factor 1 (HIF-1) signaling pathway, which is known to be crucial for intestinal stem cell function and mucosal homeostasis. Notably, blocking HIF-1 signaling abrogated the radioprotective and pro-regenerative effects of APS, indicating that APS acts through HIF-1 to foster ISC-mediated repair. These findings suggest that enhancing HIF-1-dependent pathways in ISCs is one strategy by which TCM-derived compounds can accelerate intestinal regeneration [7].

Another example of ISC modulation comes from Dendrobium species (a genus of medicinal orchids). Polysaccharides (Figure 1a) isolated from *Dendrobium fimbriatum* have demonstrated pro-regenerative effects on the intestinal epithelium by engaging ISCs’ activity via immune modulation. In an intestinal injury model, *Dendrobium fimbriatum* polysaccharide promoted the regeneration of ISCs and protected the mucosal integrity, an effect mediated by lamina propria lymphocytes (LPLs) secreting interleukin-22 (IL-22). IL-22 is a cytokine known to stimulate intestinal stem cells and progenitor cells to proliferate and differentiate, especially toward epithelial protective lineages. The co-culture of intestinal organoids with immune cells in this study provided an in vitro simulation of the gut immune microenvironment, and treatment with the *Dendrobium* polysaccharide led to the upregulation of IL-22 from LPLs, which in turn drove ISC-dependent epithelial repair. This highlights how herbal polysaccharides can indirectly activate intestinal stem cell regeneration by modulating immune cell signals in the niche (in this case, IL-22) [8,9].

Traditional multi-herb formulations can also influence stem cell-driven regeneration by shifting the balance of the epithelial lineage differentiation. Si-Ni-San (Sinisan) (Figure 1a), a classic herbal medicine compound, was recently reported to alleviate colitis injuries by promoting secretory cell lineage commitment and mucin production in colonic crypts (i.e., enhancing goblet cell differentiation). By facilitating the restoration of mucus-secreting goblet cells, Sinisan helps rebuild the protective mucus layer and supports epithelial healing [10]. This suggests that certain formulas can bias ISCs towards specific lineages (secretory cells, in this case) that are beneficial for mucosal repair. Likewise, preserving the stem cell niche is important—Paneth cells, for instance, are niche cells that support ISCs, and some Chinese herbal components may help maintain such niche factors, indirectly aiding stem cell function.

Overall, compounds like APS and Dendrobium polysaccharides illustrate a key mechanism: enhancing the intrinsic regenerative capacity of the gut via ISCs. By upregulating pathways that govern ISC proliferation, like Wnt/β-catenin or HIF-1, or by providing pro-regenerative cytokine signals (IL-22) and maintaining niche support, Chinese medicine-derived molecules can accelerate the replacement of lost or damaged intestinal epithelial cells. This ISC modulation often translates into the faster restoration of normal crypt–villus structures and improved outcomes after injury.

### 2.2. Regulation of Inflammation and Immune Modulation

Chronic or excessive inflammation in the gut can impede regeneration by causing ongoing tissue damage and by creating a hostile environment for stem cells and progenitors. Many natural products from TCM exert significant anti-inflammatory and immunomodulatory effects, which help heal the intestinal mucosa. A prominent example is berberine, an isoquinoline alkaloid from herbs like Coptis chinensis (Huanglian) [11]. Berberine has been extensively studied in models of colitis and is known to suppress key inflammatory signaling pathways, such as Nuclear Factor κB (NF-κB) and mitogen-activated protein kinases (MAPKs) [12]. In mouse colitis induced by dextran sulfate sodium (DSS), berberine (Figure 1b) treatment reduced colonic inflammation by downregulating pro-inflammatory cytokines and signaling—for instance, it inhibited the activation of NF-κB p65 and MAPK pathways in macrophages, leading to the decreased production of inflammatory mediators [13]. By dampening inflammation, berberine enabled improved crypt architecture and mucosal healing. Similarly, curcumin (from Curcuma longa turmeric, used in TCM formulations) is a well-known natural anti-inflammatory agent [14]. Curcumin can block signaling cascades like the Signal Transducer and Activator of Transcription 3 (STAT3) and the NOD-like receptor protein 3 (NLRP3) inflammasome that drive inflammation in IBD. In DSS colitis models, curcumin (Figure 1b) suppressed the phosphorylation and DNA-binding activity of STAT3 in colonic tissue, thereby reducing the downstream expression of inflammatory cytokines [15]. It also mechanistically inhibited NLRP3 inflammasome activation (by preventing upstream events like K^+^ efflux and reactive oxygen species (ROS) accumulation), which led to the lower maturation of IL-1β/IL-18 and reduced inflammatory damage [16,17]. These actions create a more permissive environment for tissue regeneration.

Beyond single compounds, multi-component herbal formulas exhibit potent immunomodulatory effects. Puerarin, an isoflavonoid from Pueraria lobata (kudzu root, often included in formulas like Gegen-Qinlian Decoction), has demonstrated both anti-inflammatory and anti-apoptotic properties in colitis [18]. In a DSS-induced colitis mouse study, puerarin (Figure 1b) markedly reduced colon inflammation by inhibiting the NF-κB pathway—treated mice had significantly lower myeloperoxidase activity and reduced levels of TNF-α, IL-1β, and IL-6 in colonic tissue. Puerarin also activated nuclear factor erythroid 2-related factor 2 (Nrf2) in the colon, which upregulated antioxidant enzymes (HO-1, SOD, etc.) and further mitigated inflammation and oxidative injury [19]. The net effect was less tissue damage and enhanced barrier preservation (discussed below). Another compound, Naringin (a flavonoid from citrus fruits, sometimes utilized in TCM) (Figure 1b), showed immune-regulating effects in a sepsis-induced intestinal injury model. Naringin-treated septic mice had significantly lower levels of TNF-α and IL-6 and higher IL-10 (an anti-inflammatory cytokine) levels, indicating a shift toward an anti-inflammatory profile. This immunomodulation was accompanied by an improved intestinal mucosal structure and higher survival rates in the Naringin group [20]. These examples underscore that many TCM-derived compounds promote intestinal regeneration indirectly by curbing inflammation—they reduce the influx of inflammatory cells and cytokines that would otherwise perpetuate the mucosal injury, thereby giving the epithelium a chance to recover.

It is also notable that some natural compounds can modulate gut immune cell populations in more specific ways. Curcumin (Figure 1b), for instance, has been reported to influence T cell differentiation in colitis, restricting pro-inflammatory Th1/Th17 responses and increasing regulatory or anti-inflammatory cell types. By altering dendritic cell and T-cell interactions, curcumin helped restore the immune balance in the gut mucosa [21]. Likewise, cannabinoids from Cannabis sativa (occasionally classed as Chinese herbal compounds in modern contexts) have immunosuppressive effects that might protect the gut during inflammation (though research is still emerging) [22]. Overall, attenuating pathological inflammation is a key mechanism by which Chinese medicine-derived compounds create conditions conducive to mucosal regeneration.

### 2.3. Improvement in Epithelial Barrier Function

The intestine is only effectively healed if the epithelial barrier—formed by tightly joined enterocytes and mucus—is re-established. A leaky or impaired barrier not only fails to absorb nutrients properly but also allows bacterial toxins and antigens to drive further inflammation, hampering regeneration. Antibiotic exposure can also precipitate profound disruptions in the gut ecosystem and intestinal mucosal barrier, evident as increased paracellular permeability and the disruption of the tight junction architecture.

Many natural compounds have been shown to strengthen the intestinal barrier by upregulating junctional proteins and enhancing mucus production. The restoration of tight junction (TJ) integrity is a common outcome observed with TCM-derived treatments. For example, in the aforementioned study of the WangshiBaochi pill (WSBCW, a Chinese herbal medicine for gastrointestinal disorders) (Figure 1c), treated mice exhibited significantly increased expression of tight junction and adherens junction proteins, such as zonula occludens-1 (ZO-1), occludins, claudins, and E-cadherin, in the intestine. Correspondingly, the WSBCW improved intestinal permeability (measured by FITC-dextran assays) in healthy mice and prevented hyper-permeability in a diarrhea model, indicating a more intact barrier. Histologically, the WSBCW increased the villus length and crypt depth, signifying enhanced mucosal structural regeneration [23]. This example illustrates how an herbal formula can promote barrier repair at the molecular level (junctional proteins) and tissue level (villus architecture).

Another example is lentinan (Figure 1c), a β-glucan from Lentinus mushrooms, which has demonstrated both preventive and therapeutic efficacy in antibiotic-induced dysbiosis. Treated mice had higher levels of butyrate and propionate in the colon, correlating with an improved epithelial energy supply and water absorption. Consequently, lentinan-treated mice exhibited strengthening of the intestinal barrier—with the colonic ZO-1 and occludin expression being restored to normal levels—and a blunted inflammatory response relative to antibiotic-only mice. Mechanistic analyses showed that lentinan attenuated NF-κB pathway activation, leading to lower concentrations of TNF-α, IL-6, and IL-1β in intestinal tissues [24].

Quercetin is a natural compound (a flavonol present in many fruits and in TCM herbs like Sophora japonica) that has strong barrier-protective effects [25]. In intestinal epithelial cell cultures, a quercetin pretreatment markedly counteracted the barrier disruption induced by injurious stimuli. Mechanistically, quercetin (Figure 1c) was shown to increase the gene and protein expression of key tight junction proteins, including ZO-1, occludin, and claudin-1, thereby maintaining the tight junction integrity. For example, in IEC-6 cell monolayers injured by indomethacin, quercetin inhibited stress-activated JNK/Src signaling that would normally downregulate TJs, and as a result, the quercetin-treated cells preserved high levels of ZO-1/occludin and avoided barrier leakage [26]. In vivo, quercetin similarly protected the intestinal barrier in a rat model of acute necrotizing pancreatitis: it prevented the loss of TJs (ZO-1, claudin-1, and occludin) and lowered markers of permeability (DAO and endotoxin) while reducing inflammatory damage [27]. These data suggest that quercetin can directly stabilize the epithelial junctional complex under stress conditions, thus supporting barrier function during regeneration.

Several other TCM-derived compounds mirror these barrier-enhancing actions. Berberine (Figure 1c) in DSS colitis models has been shown to increase the colonic expression of ZO-1, ZO-2, occludin, and E-cadherin, correlating with reduced mucosal permeability [28]. Notably, berberine also modulated the expression of genes like Zeb1 and Slug that affect the epithelial–mesenchymal transition, thereby preserving epithelial phenotypes and junctions in inflamed tissues [29,30]. Puerarin (Figure 1c) protected the integrity of TJs in ethanol-injured Caco-2 cell monolayers by influencing signaling pathways (it activated ERK1/2 MAPK and inhibited NF-κB/MLCK, which led to TJ stabilization) [31]. In DSS colitis mice, puerarin treatment increased colonic ZO-1 and occludin levels and prevented the rise in the intestinal permeability, which is consistent with a more intact barrier. Naringin (Figure 1c) was also shown to promote tight junction protein expression (ZO-1 and claudin-1) in both in vivo and in vitro sepsis models—an effect tied to its ability to inhibit the RhoA/MLCK pathway and NF-κB activation that would otherwise disrupt junctions [32]. Furthermore, compounds that increase mucus secretion or goblet cell numbers (such as Sinisan’s effect on mucin-producing cells) also improve the barrier by reinforcing the mucus layer. In summary, a critical mechanism of these natural products is reinforcing the physical epithelial barrier—by upregulating tight junctions and adherens junctions and restoring the mucus layer—which not only helps prevent further injury but also creates conditions for proper tissue regeneration and function.

### 2.4. Reduction in Oxidative Stress

Oxidative stress is a common feature of intestinal injury and inflammation, where excess ROS are generated by the immune cells and damaged epithelium. ROS and other free radicals can kill stem cells and progenitors, damage DNA, and oxidatively impair proteins and lipids, thus hindering regenerative processes. Many Chinese medicinal compounds have inherent antioxidant properties or can activate cellular antioxidant pathways, thereby mitigating oxidative stress in the gut environment and promoting healing.

A number of flavonoids in TCM, including quercetin and Icariin (from Epimedium species), are well-documented antioxidants. Quercetin (Figure 1b) not only scavenges ROS directly but also activates the Nrf2 pathway, a master regulator of antioxidant defenses in cells. In porcine enterocyte studies, quercetin prevented oxidative injury induced by the pro-oxidant diquat by activating Nrf2, which led to the upregulation of enzymes like glutathione peroxidase and superoxide dismutase. Through Nrf2, quercetin also modulated the intracellular redox state in intestinal cells and even appeared to tighten junctions in an Nrf2-dependent manner (linking antioxidant action with barrier enhancement) [33]. Puerarin has similarly been shown to elevate Nrf2 signaling and downstream antioxidant enzymes (HO-1, NQO1, catalase, etc.) in colitis mice, reducing lipid peroxidation and tissue oxidative damage [34]. By enhancing the antioxidant capacity, puerarin reduced mucosal inflammation and protected TJs, ultimately aiding in the faster recovery of the epithelium.

Herbal formulas can also reduce oxidative stress through immunomodulation. The Changyanning formula (Figure 1b), when tested in an intestinal organoid co-culture model, was found to control excessive ROS production by modulating the intestinal immune microenvironment. In that study, co-cultured immune cells produced high ROS in an untreated enteritis model; the addition of the Changyanning formula suppressed the ROS levels, likely by tempering the activation of reactive immune cells. The reduction in oxidative stress was associated with a therapeutic benefit in the enteritis model [35]. Another classical formula, Banxia Xiexin Decoction (BXD) (Figure 1b), was reported to exert anti-colitis effects by suppressing oxidative stress in the colorectum via Nrf2 activation. BXD-treated DSS colitis mice had higher colonic Nrf2 contents and lower malondialdehyde (an oxidative damage marker) levels, along with reduced inflammatory cytokines, suggesting that its amelioration of colitis was partly achieved through an antioxidant mechanism [36].

Oxidative stress and inflammation are often intertwined, and many TCM compounds address both phenomena simultaneously (e.g., curcumin’s activation of the SIRT1/Nrf2 pathway and inhibition of TLR4 in models of necrotizing colitis, which reduced both oxidative damage and inflammation). By reducing ROS levels and boosting cellular antioxidant defenses, Chinese medicine-derived compounds protect intestinal stem cells and differentiated cells from oxidative injury, thereby preserving the regenerative potential of the tissue [37]. This mechanism is especially crucial in scenarios like radiation-induced intestinal damage, where oxidative stress is a major driver of tissue injury. The radioprotective effect of Astragalus polysaccharide in irradiated intestines, for instance, is partly attributed to its antioxidant and anti-apoptotic functions, as noted in other studies. In summary, combating oxidative stress is a vital component of how these natural products foster a hospitable milieu for intestinal regeneration.

### 2.5. The Modulation of the Gut Microbiota and Its Metabolites

The gut microbiota plays an integral role in intestinal homeostasis and regeneration. Commensal bacteria produce metabolites (such as short-chain fatty acids) that nourish epithelial cells and modulate inflammation, whereas dysbiosis (an imbalanced microbiota) can perpetuate the injury by encouraging pathogen overgrowth or aberrant immune activation. One reason is the overuse of antibiotics. Broad-spectrum antibiotics sharply reduce the diversity of commensal microbiota and deplete key SCFA-producing bacteria. In antibiotic-treated mice, colonic concentrations of acetate, propionate, and butyrate drop to roughly half of the normal levels [38]. Fascinatingly, many Chinese herbal compounds and formulas can reshape the gut microbiota composition, and this microbiota modulation is increasingly recognized as a mechanism contributing to their therapeutic effects on the intestine.

For example, the WSBCW (Figure 1d) herbal preparation mentioned earlier not only acted directly on the host (improving junctions and reducing cytokines) but also significantly altered the gut microbiota profile of mice. The WSBCW treatment increased the relative abundance of beneficial genera like Bifidobacterium (often associated with gut health) and Desulfovibrio while suppressing the overgrowth of potentially harmful bacteria like the Bacteroides fragilis group [23]. These changes in flora were correlated with improved intestinal function and reduced diarrhea in the treated animals. Similarly, berberine has well-documented microbiota effects: it exhibits broad antimicrobial activity that can reduce pathogens, but it tends to favor an increase in short-chain fatty acid-producing bacteria at sub-antimicrobial doses. One study found that the berberine treatment led to metabolic changes, as seen in healthy volunteers’ stool samples, consistent with an increase in the butyrate production pathway [39]. In animal models, berberine has been shown to restore a dysbiotic microbiota toward a more balanced state in colitis—for instance, in DSS-treated cats, berberine increased levels of beneficial gut bacteria and improved the overall microbial diversity [40]. By promoting butyrate and other SCFA productions, berberine can indirectly support the intestinal epithelial energy supply and enhance mucosal repair.

Cereus sinensis polysaccharide (CSP-1) (Figure 1d)—a purified natural polysaccharide—helped reverse antibiotic-induced microbial imbalances. In a mouse model of antibiotic-associated diarrhea (AAD), CSP-1 enriched SCFA-producing genera (e.g., Phascolarctobacterium and Bifidobacterium); restored microbial diversity; and elevated cecal acetate, propionate, and butyrate to normal levels. In line with this, CSP-1-treated mice showed reduced intestinal fluid loss and improved weight gain compared to untreated AAD controls, as well as a significantly dampened secretion of inflammatory cytokines (TNF-α, IL-1β, and IL-2) back to baseline levels [41]. Beyond that, a polysaccharide from Chinese yam (Figure 1d) significantly alleviated AAD in mice by upregulating the abundance of probiotic bacteria, suppressing the overgrowth of opportunists, and increasing cecal SCFA levels [42].

Complex herbal formulas often have even more pronounced microbiota-modulating capabilities. A striking example is Gegen-Qinlian Decoction (GQD) (Figure 1d), which contains herbs like Pueraria (gegen) and Coptis (which has berberine). In a UC rat model, a modified GQD significantly raised the abundance of Akkermansia and Romboutsia—both genera associated with mucosal health—and decreased Escherichia–Shigella populations that include pathogenic bacteria [43]. These microbiota changes were accompanied by the increased production of short-chain fatty acids (acetate, propionate, and butyrate) in the colon and an improvement in colonic inflammation and histology. SCFAs like butyrate are known to fuel colonocytes and promote ISC differentiation into the functional epithelium; they also have anti-inflammatory properties [44]. Thus, GQD’s ability to foster SCFA-producing commensals and reduce opportunistic bacteria likely created a biochemical environment favorable for regeneration (e.g., through butyrate-mediated enhancement of epithelial barrier and regulatory immune responses). In diarrheic piglet models, GQD similarly increased beneficial commensals (e.g., Lactobacillus and Ruminococcus) and SCFA levels while inhibiting pathogenic E. coli, leading to the restoration of the mucosal structure and function [45].

Curcumin (Figure 1d) has also been shown to selectively modulate the gut flora composition. In one study, curcumin supplementation in colitis mice helped restore a healthier microbiome, which was linked to reduced inflammation and even improvements in gut–brain-axis-related symptoms [46]. New formulations like nanoparticle-encapsulated curcumin appear to enhance this effect; a recent report noted that a nano-curcumin greatly alleviated experimental colitis by shifting the gut microbiota structure and influencing mucosal immune cell regulation [47]. This suggests that curcumin’s benefits are partly microbiota-mediated, as microbiome changes can lead to secondary effects like altered bile acid profiles or immune modulation.

In summary, microbiota interactions are a crucial mechanism through which Chinese medicine-derived compounds exert regenerative effects. By pruning the overgrowth of pathogenic bacteria and enriching commensal strains that produce beneficial metabolites (SCFAs, etc.), these interventions reduce mucosal insults (e.g., less toxins and pathogen-associated inflammation) and increase factors that promote healing (e.g., butyrate to enhance barrier function and ISC differentiation). The result is a more balanced microbial ecosystem that supports the intestinal epithelium’s recovery. This multi-hit approach—simultaneously targeting host pathways and microbial communities—underscores the holistic nature of TCM therapeutics in intestinal health.

## 3. Experimental Evidence from In Vitro and In Vivo Studies

### 3.1. In Vitro Models: Intestinal Organoids and Cell Cultures

Advanced in vitro models have been instrumental in elucidating how Chinese medicinal compounds act on intestinal cells and stem cells. Intestinal organoids (Figure 2a)—three-dimensional mini-gut structures grown from ISCs—provide a particularly powerful platform to study regeneration in a controlled setting. Researchers have utilized organoid cultures to directly observe the pro-regenerative actions of TCM-derived compounds. For example, the effect of Dendrobium polysaccharides on ISC regeneration was demonstrated using a co-culture of mouse intestinal organoids with immune cells. In this system, organoids (containing stem cells and epithelium) normally suffered damage when exposed to injurious stimuli, but treatment with the Dendrobium fimbriatum polysaccharide, in the presence of LPL immune cells, led to significantly improved organoid growth and recovery [9]. The IL-22 secretion from immune cells (stimulated by the polysaccharide) was identified as the key factor driving organoid (and ISC) regeneration. This organoid co-culture approach closely mimics the in vivo intestinal microenvironment and clearly showed that a TCM compound can activate intestinal stem cell-mediated repair via immune modulation.

In another study, organoids were used to evaluate the Astragalus polysaccharide (APS). Ding et al. established intestinal organoid cultures from mice and found that APSs markedly enhanced organoid-forming efficiency and growth after radiation damage. Essentially, APS-treated organoids had higher survival and regeneration of crypt-like domains, aligning with the compound’s stimulation of ISC proliferation. These ex vivo organoid results reinforced the in vivo findings that the APS promotes ISC-driven regeneration through HIF-1 signaling [7]. Organoid assays thus confirmed that the APS has a direct effect on epithelial stem cells’ capacity to regrow tissue.

Aside from organoids, traditional cell culture models have provided insights into barrier-related mechanisms. Caco-2 and IEC-6 cell monolayers (Figure 2a) (which model intestinal epithelial layers) have been treated with compounds like quercetin, puerarin, and berberine to test their ability to prevent barrier disruption. Quercetin’s ability to preserve tight junctions under chemical stress was demonstrated in IEC-6 cells, as described earlier [25]. Puerarin prevented ethanol-induced tight junction dissociation in Caco-2 cells by modulating NF-κB and MAPK signaling, maintaining high levels of ZO-1 and occludin in the cell junctions [30]. Berberine protected Caco-2/HT29 co-culture barriers from inflammatory (TNF-α) injuries, helping maintain E-cadherin and other junctions dose-dependently [48]. These cell culture experiments allow the precise dissection of pathways—for instance, using siRNA knockdowns or pathway inhibitors, researchers have verified that blocking Nrf2 or inhibiting certain kinases will negate the barrier-protective effects of these compounds. Such findings strongly indicate that the compounds act on specific epithelial cell signaling cascades (e.g., quercetin on JNK/Src and Nrf2, puerarin on NF-κB/MLCK) to exert their pro-regenerative, barrier-stabilizing outcomes.

Another interesting in vitro system is organ-on-a-chip technology (Figure 2a), though its use in TCM research is still in its early stages. To date, few organ-on-chip studies have been performed for TCM and the gut; one report examined the potential nephrotoxicity of a plant compound (kaempferol) on a kidney chip and found no damage at relevant doses, hinting at the value of such models for safety [49]. Gut-on-chip models may be applied to study dynamic interactions (peristalsis and flow) with herbal compounds. Organoids are currently the primary in vitro tool demonstrating the intestinal regenerative effects of Chinese herbal compounds, complemented by simpler cell cultures for mechanistic probing.

In summary, in vitro experiments (particularly organoids) have provided compelling evidence that Chinese medicine-derived compounds can directly act on intestinal epithelial systems to promote regeneration. They allow the observation of stem cell proliferation, crypt-like domain formation, differentiation, and barrier integrity in response to these natural products, under controlled conditions free from systemic influences. These studies validate that these compounds have intrinsic efficacy on gut tissues, thus substantiating their observed potential in whole-animal models.

### 3.2. In Vivo Studies: Animal Models of Intestinal Injury and Disease

A wealth of in vivo evidence from animal studies supports the regenerative effects of Chinese medicinal compounds on the intestine. These studies span various models—from chemically induced colitis and diarrhea models to radiation injury and sepsis—and consistently show improved outcomes in animals treated with natural compounds or herbal formulas.

One of the most common models, DSS-induced colitis (Figure 2b) in mice or rats, simulates ulcerative colitis with epithelial erosion and ulceration. Multiple compounds have been tested in this model. Berberine-treated colitis mice exhibit significantly milder disease, showing reduced weight loss and diarrhea, longer colon lengths, and lower histopathology scores compared to untreated controls. At the cellular level, berberine reduces neutrophil infiltration and preserves the crypt architecture; molecularly, it inhibits colonic NF-κB activation and upregulates antioxidant enzymes (consistent with its anti-inflammatory/antioxidant actions noted earlier) [35]. Curcumin has even progressed to clinical trials after showing promising animal data—in DSS colitis mice, it ameliorated inflammation (via STAT3/NLRP3 suppression) [15], and a randomized controlled trial involving ulcerative colitis patients showed that oral curcumin (1500 mg daily) improved clinical remission rates and quality of life while lowering inflammatory markers (C-reactive protein, etc.) [50]. This translation from animals to humans highlights the therapeutic potential of such compounds for enhancing mucosal healing in IBD.

Classic TCM formulas have also been validated in vivo. BXD was found to significantly attenuate colitis in DSS-treated mice, improving body weight recovery and reducing colon damage. Its mechanism involved blocking pro-inflammatory NF-κB signaling and boosting Nrf2-driven antioxidant responses in colonic tissue [35]. As a result, BXD-treated mice had less crypt loss and better preservation of the mucosal barrier than the controls. Si-Ni-San (Sinisan), mentioned earlier for promoting goblet cell regeneration, has also shown efficacy in a rat colitis model by restoring mucosal goblet cell numbers and mucin levels, leading to an improved stool form and reduced inflammatory ulceration [10]. GQD, besides modulating the microbiota as described, ameliorated the pathology in UC mice—reducing diarrhea and bleeding—through downregulating hyperactive signaling pathways like PI3K/Akt and EGFR that contribute to epithelial injury [51].

In models of intestinal dysmotility and diarrhea, such as IBS-like (Figure 2b) or infective diarrhea models, TCM compounds also enhance gut structure recovery. The herbal pill WSBCW improved a castor oil-induced diarrhea model in mice, not only by firming up stools but by increasing the villus height and crypt depth in the small intestine, indicating enhanced epithelial regeneration post-insult. It also upregulated tight junctions and reduced plasma cytokines in these mice, demonstrating both barrier and immune improvements. These changes translated into better gastrointestinal functional parameters: WSBCW-treated animals experienced faster restoration of normal gut transit and permeability than the controls [23]. Another formula, Sishen Wan, was shown to protect the intestinal barrier in a diarrhea-predominant IBS model by reducing endoplasmic reticulum stress in enterocytes, thereby improving tight junction integrity and relieving symptoms [52].

Another noteworthy example is the AAD model (Figure 2b). A growing body of research indicates that natural compounds—including those derived from Traditional Chinese Medicine—can be used to prevent or ameliorate AAD. These natural compounds restore a healthy microbiota, strengthen the intestinal barrier, and modulate immune responses. Lizhong-Tang—composed of ginseng, Atractylodes, ginger, and licorice—demonstrated potent anti-AAD effects in a lincomycin-induced mouse model. Lizhong-Tang-treated mice had markedly lower diarrhea scores and maintained their body weight better than untreated animals. Microbiome analyses (T-RFLP and 16S sequencing) confirmed that Lizhong-Tang corrected antibiotic-induced shifts in the gut microbiota composition, helping to ameliorate dysbiosis to achieve a healthier profile [53]. Berberine exhibits broad antimicrobial activity in the gut and favorably reshapes the microbiota composition. In the context of antibiotic-associated C. difficile infections, berberine and the flavone baicalein (from Scutellaria baicalensis) have been shown to inhibit the growth and virulence of C. difficile. These compounds also downregulated the expression of toxin synthesis genes in C. difficile, thereby attenuating the pathogen’s cytotoxic effects. By curbing opportunistic pathogens and their toxins, such phytochemicals can prevent the exacerbation of antibiotic-associated colitis and contribute to preserving the epithelial barrier [54].

Severe injury models provide further evidence. In radiation-induced intestinal syndrome (Figure 2b), where massive ISC loss leads to lethal GI tract failure, the APS from Astragalus dramatically improved outcomes: APS-treated mice had a higher survival rate after high-dose irradiation, the preservation of crypts, and the rapid re-epithelialization of the intestine. Mice in the APS group showed robust ISC recovery and proliferating cell nuclear antigen (PCNA) staining in regenerating crypts compared to the irradiated controls, aligning with APS’s HIF-1-mediated stem cell protection [7]. This suggests that some TCM compounds might serve as radioprotectants to enhance post-injury regeneration when conventional options (like cytokine therapies) are limited. In polymicrobial sepsis (which can cause mucosal degeneration), Naringin significantly improved survival in a cecal puncture model by preserving the gut barrier and modulating cytokines. Naringin-treated septic mice had notably less intestinal damage and lower bacterial translocation, likely because their tight junctions were maintained and the inflammatory injury was controlled [20].

Across these diverse models, a common theme emerges: animals receiving Chinese medicine-derived compounds or formulas consistently show faster and more complete regeneration of the intestinal lining after injury. This is evidenced by metrics like the increased villus length, deeper crypts with abundant goblet and absorptive cells, lower injury scores, and restored barrier function tests (e.g., permeability assays). The mechanistic hallmarks—reduced inflammatory infiltrates, higher ISC counts (Lgr5^+^ cells), greater tight junction protein staining, and lower oxidative damage—corroborate the modes of action discussed in the previous section.

It is worth noting that, in some cases, these compounds are moving toward clinical evaluation. Curcumin’s positive trial in ulcerative colitis and accumulating clinical observations that TCM formulations (e.g., herbal combinations like Qing-dai or Plantain extract) can induce mucosal healing in IBD contribute to an optimistic belief that the preclinical findings are translatable. However, clinical evidence is still relatively sparse, and further trials are needed for most compounds. It is crucial to address factors such as the bioavailability (some compounds like curcumin are poorly absorbed unless formulated specifically) and quality control of herbal extracts.

In summary, in vivo studies strongly support the efficacy of Chinese medicine-derived natural compounds in promoting intestinal regeneration in various contexts of disease and injury. They validate that these compounds’ multifaceted mechanisms—anti-inflammatory, barrier-enhancing, antioxidant, microbiome-modulating, and stem cell-activating effects—indeed culminate in the tangible healing of the gut lining in living organisms. These findings lay a strong foundation for clinical trials and eventually integrating such natural therapies into the management of gastrointestinal disorders where regeneration is needed.

## 4. Translational Challenges of Natural Compounds in Intestinal Therapy: Safety, Standardization, and Bioavailability

Natural compounds such as curcumin, berberine, ginsenosides, plant polysaccharides, and quercetin show therapeutic promise for intestinal repair, yet each one faces specific translational hurdles. Table 1 presents the key pathways, evidence basis, and clinical progress of major natural compounds. Curcumin is generally safe, but rare hepatotoxicity and potential drug interactions (e.g., with anticoagulants) warrant caution. Its extreme variability in turmeric preparations complicates standardization, and its poor oral bioavailability—due to rapid metabolism—requires enhancers like piperine or nanoformulations [55]. Berberine has a favorable safety profile but is contraindicated in pregnancy and may interact with drugs via CYP3A4 or P-glycoprotein inhibition. Its defined structure allows for easier standardization, but poor systemic absorption and high first-pass metabolism demand high doses or advanced delivery systems [56]. Ginsenosides, though typically safe, may affect sleep and interfere with warfarin or immunosuppressants. Variability in species, processing, and compound ratios hinders standardization. Most ginsenosides have low absorption, relying on the gut microbiota to generate bioactive metabolites like Compound K [57]. Polysaccharides (e.g., from Astragalus or mushrooms) are safe and act locally via prebiotic effects and immune modulation, but their large size precludes systemic absorption. Their complex, heterogeneous structures pose major standardization challenges; batch-to-batch consistency is difficult without advanced analytical techniques [58,59]. Quercetin, a widely studied flavonoid, is safe at moderate doses but can affect cytochrome enzymes and drug levels. While its purified form simplifies standardization, its bioavailability is limited by its poor solubility and metabolism [60].

**Table 1 biomolecules-15-01212-t001:** A summary of the major natural compounds reviewed, including their proposed molecular targets, evidence base, and status in clinical testing.

Compound	Key Pathways/Actions	Evidence Level	Clinical Trial Status
Curcumin (turmeric)	Inhibits NF-κB, TLR4/AP-1, and pro-inflammatory cytokines; activates Nrf2 antioxidant response; modulates gut microbiome (promotes SCFA-producing flora).	Extensive preclinical evidence (in vitro and animal models of colitis); multiple clinical trials in IBD and other GI disorders.	Completed: Several RCTs in ulcerative colitis showed symptom improvement and remission maintenance (approved for use as adjunct in IBD in some settings; further trials ongoing [61]).
Quercetin (flavonoid)	Antioxidant (scavenges ROS and upregulates HO-1 via Nrf2); suppresses NF-κB and MAPK signaling (reduces IL-8 and TNF-α); stabilizes mast cells; helps restore tight junction proteins in gut epithelium.	Preclinical evidence from studies in vitro (cell protection) and in rodents (DSS colitis models); some human data (dietary intake correlational studies and small pilot trials).	Completed: A small RCT in ulcerative colitis reported reduced disease activity and inflammation with quercetin supplementation [62]. Ongoing: Further clinical studies in IBD and IBS are in early phases (exploratory).
Berberine (Coptis alkaloid)	Activates AMPK; inhibits NF-κB/STAT3 and inflammasome pathways (IL-1β, IL-6, and TNF-α); modulates gut microbiota (antibacterial against harmful flora; promotes short-chain fatty acid producers); may enhance Wnt/β-catenin signaling for mucosal repair.	Strong preclinical evidence (multiple animal colitis studies showing improved mucosal healing); numerous clinical trials for metabolic and GI outcomes (diabetes, H. pylori, and diarrhea).	Completed: RCTs in ulcerative colitis and chronic diarrhea/IBS have shown benefit (e.g., berberine improved histological scores in UC [63]). Ongoing: Trials in IBD (phase II in China) and colon polyp prevention are in progress.
Ginsenosides (Panax ginseng)	Multi-target immunomodulation. Attenuate NF-κB and MAPK signaling (inflammatory cytokines IL-1β and IL-17); some ginsenosides (Rg1 and Rb1) activate Wnt/β-catenin and EGFR pathways supporting epithelial regeneration. Antioxidative effects (reduce ROS, upregulate SOD, etc.); promote balance of Th17/Treg in gut.	Preclinical evidence only: in vitro studies on immune cells; several mouse colitis models with isolated ginsenosides (Rb1, Rg1, Rk3, etc.) showing reduced inflammation and faster mucosal healing [64,65].	Not yet tested clinically for intestinal injury. (Ginseng extracts have been clinically studied for fatigue, metabolism, etc., but no clinical trials to date have focused on intestinal regeneration/IBD. Currently used as a supplement in humans and is not an established therapy for gut disease.)
Astragalus Polysaccharide (APS)	Enhances intestinal stem cell (ISC) activation via HIF-1α signaling; promotes crypt cell survival and proliferation; immunomodulatory—reduces TNF-α, IL-1β, and IFN-γ levels in injured gut; regulates gut microbiota and increases SCFA production (butyrate), leading to improved tight junction integrity.	Preclinical evidence: robust evidence in animal models of intestinal injury (radiation enteropathy and DSS colitis [7]); organoid culture studies confirming ISC regeneration effects.	Not yet tested clinically. (Experimental stage—no human trials so far. Astragalus-based TCM remedies are used empirically for GI health, but APS as an isolated compound has not yet undergone clinical trials.)
Lentinan (shiitake β-glucan)	Stimulates immune cells (macrophages and dendritic cells) via Dectin-1 and other pattern recognition receptors, inducing cytokines (IL-2 and TNF-α) that aid in tissue repair; may enhance IgA secretion and mucosal immunity; antioxidant and anti-apoptotic effects on epithelial cells reported. Indirectly supports gut barrier by fostering a protective immune microenvironment.	Preclinical evidence: in vitro immunological studies; animal models of infection and cancer (shows gut protection as side observation). In GI injury models, data are sparse, but related mushroom polysaccharides show decreased inflammation and oxidative damage.	Completed (other indications): Used clinically as an injected cancer adjuvant (extensive experience in humans for oncology). Not formally trialed for IBD or mucosal healing. (Only anecdotal or compassionate use in GI disorders; no RCTs in intestinal regeneration.)
CSP-1 (sulfated polysaccharide fraction)	Anti-inflammatory and prebiotic: inhibits TLR4/MyD88/NF-κB signaling cascade, leading to reduced NF-κB activation; downregulates IL-23/IL-17 axis (fewer Th17 pro-inflammatory cells); helps restore tight junction proteins and barrier function (likely via its sulfate groups, binding gut mucins and modulating microbiota).	Preclinical evidence: early-stage studies in cell culture and rodent antibiotic-associated diarrhea models. Shown to stabilize gut barrier and lower inflammatory mediators in these models [59].	Not yet tested clinically. (Research compound only—no human studies. Needs further animal validation before any clinical trials.)

Abbreviations: NF-κB—nuclear factor κB; TLR4—toll-like receptor 4; AP-1—activator protein 1; Nrf2—nuclear factor erythroid 2-related factor 2; ROS—reactive oxygen species; AMPK—AMP-activated protein kinase; STAT3—signal transducer and activator of transcription 3; Wnt—Wnt/β-catenin pathway; HIF-1—hypoxia-inducible factor 1; ISC—intestinal stem cell; SCFA—short-chain fatty acid; Th17—T helper 17 cells; DSS—dextran sulfate sodium; IBD—inflammatory bowel disease; and RCT—randomized controlled trial.

In summary, while these compounds have demonstrated local and systemic benefits in preclinical models, their clinical translation is constrained by an inconsistent quality, poor pharmacokinetics, and potential interactions. Overcoming these challenges—through better drug delivery systems, rigorous clinical trials with standardized preparations, and a deeper understanding of herb–drug interactions—will be key to integrating natural compounds into mainstream therapies for intestinal regeneration.

## 5. Conclusions

Natural compounds derived from Traditional Chinese Medicine exhibit remarkable capacity to promote intestinal regeneration through multiple converging mechanisms. As detailed above, these compounds can stimulate intestinal stem cells, helping to replenish the epithelium after injury, and they can favor the differentiation of protective cell lineages (such as goblet and Paneth cells) that support mucosal healing. At the same time, they exert anti-inflammatory effects, dampening cytokine storms and immune overactivation that impede healing in conditions like IBD. In addition to reducing inflammation, many of these agents strengthen the gut’s barrier function—by upregulating tight junction proteins and restoring the integrity of the mucosal layer—which prevents further damage and sets the stage for proper tissue recovery. Additionally, their antioxidant actions protect vulnerable stem and progenitor cells from oxidative injury, ensuring the regenerative machinery of the gut can function optimally even in stress conditions. Finally, by modulating the gut microbiota imbalance caused by antibiotics and other factors, Chinese medicine-derived compounds help re-establish a symbiotic microbial environment that produces beneficial metabolites (such as SCFAs) and curbs pathogenic insults. This holistic, multi-target approach is a signature feature of TCM therapies and is ideally suited to a complex process like intestinal regeneration, which involves an interplay between epithelial cells, immune cells, and microbiota.

The experimental evidence supporting these benefits is extensive. From controlled in vitro studies with intestinal organoids to a variety of animal models of intestinal injury, TCM compounds have consistently demonstrated enhanced mucosal healing—with outcomes such as the faster regeneration of villi/crypts, improved histological scores, reduced leakiness, and higher survival in severe injury models. Notably, some compounds like curcumin have translated this promise into early clinical success, indicating that preclinical mechanisms manifest in human disease to some extent. Herbal formulas, which combine multiple active ingredients, often show synergistic effects on multiple pathways (for example, simultaneously reducing NF-κB-driven inflammation and increasing Nrf2-driven antioxidant responses). This multi-component synergy might be crucial for tackling complex diseases like IBD where no single target is sufficient.

Although these mechanisms are promising and some natural compounds have made clinical progress, the clinical translation of natural compounds still faces significant challenges, including safety concerns, standardization issues, and poor bioavailability. For example, the hepatotoxic risk and drug interactions of curcumin, the absorption limitation of berberine, and the structural complexity of polysaccharides all require targeted solutions such as nanoformulations or advanced delivery systems. Standardization and rigorous clinical trials of herbal extracts are essential to bridge the gap between preclinical efficacy and clinical applicability.

These findings encourage further scientific and clinical exploration of Chinese medicine-derived compounds as therapeutic agents for intestinal repair. There is a need for well-designed clinical trials to evaluate efficacy and safety in patients (for example, in ulcerative colitis remission or in mitigating the gastrointestinal side effects of radiation/chemotherapy). Issues of standardization, dosage, and bioavailability must be addressed, given that natural extracts can vary in composition. It is also important to pinpoint biomarkers of response—for instance, identifying which subset of patients (perhaps defined by the microbiome profile or genetic markers) might benefit most from a given herbal treatment. Modern techniques, including metabolomics and advanced imaging, could help in understanding exactly how these compounds interact with the gut ecosystem in humans.

In conclusion, Chinese medicine-derived natural compounds represent a promising and mechanistically rich source of therapies for enhancing intestinal regeneration. They embody a multi-modal strategy: protecting and activating stem cells, tempering the immune response, fortifying the barrier, quenching oxidative stress, and cultivating a healing-friendly microbiome. Such a comprehensive approach is highly relevant for diseases characterized by epithelial injury and defective healing. By bridging traditional knowledge with modern biomedical research, we can potentially develop novel treatments that safely and effectively promote the restoration of the gut integrity and function in patients suffering from inflammatory and injurious gastrointestinal conditions.

## Figures and Tables

**Figure 1 biomolecules-15-01212-f001:**
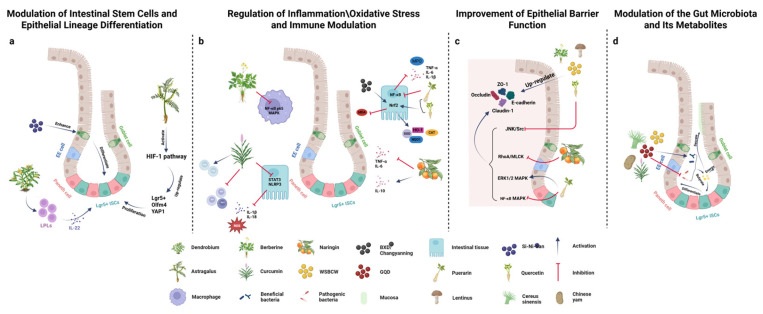
Mechanisms of action of Chinese medicine-derived natural compounds in intestinal regeneration. This schematic illustrates four key mechanisms by which natural compounds promote intestinal regeneration, integrating molecular, cellular, and microbial interactions to depict the multifaceted roles of natural compounds in intestinal repair and homeostasis. These include (**a**) the regulation of the intestinal stem cell and epithelial cell lineage differentiation; (**b**) the modulation of inflammation/oxidative stress and immune regulation; (**c**) the improvement of the epithelial barrier function; and (**d**) the modulation of the intestinal microbiota and its metabolites.

**Figure 2 biomolecules-15-01212-f002:**
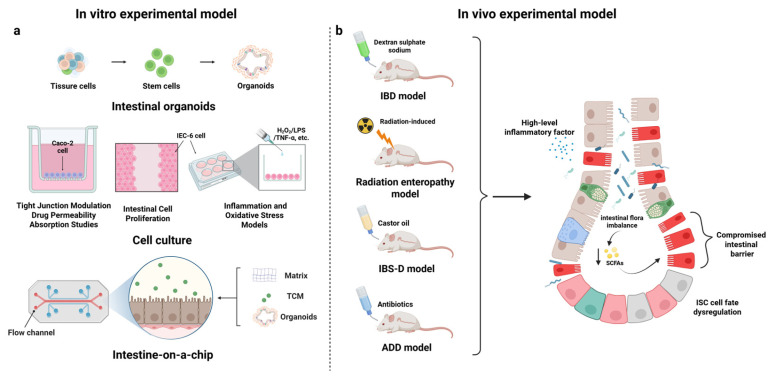
In vitro and in vivo intestinal experimental models for natural compounds derived from Chinese medicine. This infographic provides an overview of experimental models used to study intestinal regeneration and disease mechanisms, integrating in vitro and in vivo approaches. In vitro models utilize (**a**) intestinal organoids, intestinal epithelial mimic cells (Caco-2 and HT29), and intestinal organ chips to conduct drug permeability and absorption studies and assess intestinal barrier integrity, cell proliferation, inflammation (e.g., TNF-α), and oxidative stress. In vivo models focus on chemical and physical induction, including (**b**) inflammatory bowel disease (IBD) induced by dextran sulfate sodium (DSS), intestinal damage induced by ionizing radiation (e.g., γ rays and X rays), diarrhea-predominant irritable bowel syndrome (IBS-D) induced by castor oil gavage, and antibiotic-associated diarrhea (AAD) induced by antibiotics. These models comprehensively simulate intestinal inflammation, microbial imbalances, epithelial barrier loss, and intestinal stem cell dysregulation.

## Data Availability

Not applicable.

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
