# Peer review of "Chinese Medicine-Derived Natural Compounds and Intestinal Regeneration: Mechanisms and Experimental Evidence"

_biomolecules, 2025, doi:10.3390/biom15091212_

Round 1

Reviewer 1 Report

Comments and Suggestions for Authors

I read comprehensive review on "Chinese medicine-derived natural compounds and intestinal regeneration" with great interest. The authors have provided valuable insights into the multifaceted mechanisms by which these bioactive compounds promote intestinal homeostasis.

However, the current review would be significantly strengthened by adding a dedicated section on "Antibiotic-Associated Diarrhea Models and Natural Compounds" which should include:
- Mechanisms of antibiotic-induced gut microbiota dysbiosis and intestinal barrier dysfunction
- Preventive and therapeutic effects of natural compounds against antibiotic-associated diarrhea.

Author Response

Comment1:

I read comprehensive review on "Chinese medicine-derived natural compounds and intestinal regeneration" with great interest. The authors have provided valuable insights into the multifaceted mechanisms by which these bioactive compounds promote intestinal homeostasis.

However, the current review would be significantly strengthened by adding a dedicated section on "Antibiotic-Associated Diarrhea Models and Natural Compounds" which should include:
- Mechanisms of antibiotic-induced gut microbiota dysbiosis and intestinal barrier dysfunction 
- Preventive and therapeutic effects of natural compounds against antibiotic-associated diarrhea.

Response1: Thank you for pointing this out. We agree with this comment.Therefore, we have revised the manuscript to include the following additions: (1) Seciton 2.3: Improvement of Epithelial Barrier Function, we incorporated the concept of intestinal and mucosal barrier damage induced by antibiotic exposure, along with an example of barrier repair by natural compounds. (2) Section 2.5: Modulation of the Gut Microbiota and Its Metabolites, we added the concept of antibiotic-induced gut microbiota imbalance and provided an example of natural compounds restoring microbial diversity in such models. (3) Section 3.2: In Vivo Studies—Animal Models of Intestinal Injury and Disease, we included content on the antibiotic-induced diarrhea model. These are already marked in red.

Reviewer 2 Report

Comments and Suggestions for Authors

The manuscript offers a thorough and well-organized synthesis of current evidence on Chinese medicine–derived natural compounds in intestinal regeneration, supported by both mechanistic discussion and experimental data. The clear subdivision into mechanisms—ranging from intestinal stem cell activation to microbiota modulation—enhances readability and reflects a deep understanding of the multifactorial nature of mucosal repair. The breadth of in vitro and in vivo evidence cited is commendable, and the inclusion of specific compounds and formulations with corresponding mechanistic pathways makes the review highly informative. The schematic figures and integration of microbiota-related findings further strengthen the translational relevance of the work.

That said, some sections would benefit from additional critical appraisal. While the mechanistic overviews are detailed, the narrative could more explicitly address the strength and limitations of existing evidence, particularly in distinguishing between preclinical findings and clinical applicability. Expanding on potential safety concerns, standardization challenges, and bioavailability issues for certain compounds (e.g., curcumin, berberine) would provide a more balanced perspective. Furthermore, the review might be strengthened by synthesizing the discussed mechanisms into a concise table mapping each compound to its key pathways, evidence level, and clinical trial status. This would improve accessibility for readers seeking quick reference points.

Overall, the review is comprehensive and timely, with a logical structure and strong literature support. Minor refinements to highlight evidence quality, potential research gaps, and translational hurdles would further enhance its impact and practical utility for researchers and clinicians in the field.

Author Response

Comment1:

The manuscript offers a thorough and well-organized synthesis of current evidence on Chinese medicine–derived natural compounds in intestinal regeneration, supported by both mechanistic discussion and experimental data. The clear subdivision into mechanisms—ranging from intestinal stem cell activation to microbiota modulation—enhances readability and reflects a deep understanding of the multifactorial nature of mucosal repair. The breadth of in vitro and in vivo evidence cited is commendable, and the inclusion of specific compounds and formulations with corresponding mechanistic pathways makes the review highly informative. The schematic figures and integration of microbiota-related findings further strengthen the translational relevance of the work.

That said, some sections would benefit from additional critical appraisal. While the mechanistic overviews are detailed, the narrative could more explicitly address the strength and limitations of existing evidence, particularly in distinguishing between preclinical findings and clinical applicability. Expanding on potential safety concerns, standardization challenges, and bioavailability issues for certain compounds (e.g., curcumin, berberine) would provide a more balanced perspective. Furthermore, the review might be strengthened by synthesizing the discussed mechanisms into a concise table mapping each compound to its key pathways, evidence level, and clinical trial status. This would improve accessibility for readers seeking quick reference points.

Overall, the review is comprehensive and timely, with a logical structure and strong literature support. Minor refinements to highlight evidence quality, potential research gaps, and translational hurdles would further enhance its impact and practical utility for researchers and clinicians in the field.

Response1:Thank you for pointing this out. We agree with this comment.Therefore, We have revised the manuscript to include an additional section, Section 4: Translational Challenges of Natural Compounds in Intestinal Therapy—Safety, Standardization, and Bioavailability. In this section, we also added a table summarizing the molecular targets, supporting evidence, and clinical trial status of major natural compounds. These are already marked in red